# Comparison of Lake Area Extraction Algorithms in Qinghai Tibet Plateau Leveraging Google Earth Engine and Landsat-9 Data

**Xusheng Li** [1], **Donghui Zhang** [2], **Chenchen Jiang** [3], **Yingjun Zhao** [1,*], **Hu Li** [4], **Donghua Lu** [1], **Kai Qin** [1], **Donghua Chen** [5], **Yufeng Liu** [5], **Yu Sun** [1] **and Saisai Liu** [5]

1   National Key Laboratory of Remote Sensing Information and Imagery Analyzing Technology, Beijing Research Institute of Uranium Geology, Beijing 100029, China
2   National Engineering Laboratory for Satellite Remote Sensing Application (NELRS), Aerospace Information Research Institute, Chinese Academy of Sciences, Beijing 100094, China
3   Institute of Remote Sensing and Geographic Information System, School of Earth and Space Sciences, Peking University, Beijing 100871, China
4   School of Geography and Tourism, Anhui Normal University, Wuhu 241000, China
5   School of Computer and Information Engineering, Chuzhou University, Chuzhou 239000, China
*   Correspondence: zhaoyingjun@briug.cn; Tel.: +86-010-64964719

**Abstract:** Monitoring the lake waterbody area in the Qinghai–Tibet Plateau (QTP) is significant in dealing with global climate change. The latest released Landsat-9 data, which has higher radiation resolution and can be complemented with other Landsat data to improve imaging temporal resolution, have great potential for applications in lake area extraction. However, no study is published on identifying waterbodies and lakes in large-scale plateau scenes based on Landsat-9 data. Therefore, we relied on the Google Earth Engine (GEE) platform and selected ten waterbody extraction algorithms to evaluate the quantitative evaluation of waterbody and lake area extraction results on the QTP and explore the usability of Landsat-9 images in the relationship between the extraction accuracy and the algorithm. The results show that the random forest (RF) algorithm performs best in all models. The overall accuracy of waterbody extraction is 95.84%, and the average lake waterbody area extraction error is 1.505%. Among the traditional threshold segmentation waterbody extraction algorithms, the overall accuracy of the NDWI waterbody extraction method is 89.89%, and the average error of lake waterbody area extraction is 3.501%, which is the highest performance model in this kind of algorithm. The linear regression coefficients of NDVI and reflectance of Landsat-8 and Landsat-9 data are close to 1, and $R^2$ is more significant than 0.91. At the same time, the overall accuracy difference of water extraction between the two data is not better than 1.1%. This study proves that Landsat-9 and Landsat-8 data have great consistency, which can be used for collaborative analysis to identify plateau waterbodies more efficiently. With the development of cloud computing technologies, such as Gee, more complex models, such as RF, can be selected to improve the extraction accuracy of the waterbody and lake area in large-scale research.

**Keywords:** Landsat-9 data; Qinghai–Tibet Plateau; lake waterbody; GEE; algorithms comparison

## 1. Introduction

The Qinghai–Tibet Plateau (QTP), referred to as the 'water tower of Asia', is the birthplace of many major rivers, such as the Yangtze River and the Yellow River. It is about 2.6 million km$^2$ in area, most of which lies at an elevation of more than 4 km above sea level [1]. The unique alpine terrain of the QTP blocked and raised the warm and humid South Asian monsoon [2], forming rich water resource reserves in the region. The QTP contains approximately 1400 lakes of an area larger than 1 km$^2$, with a total area of about 50,000 km$^2$. The area of lakes on the QTP affected by runoff and precipitation reaches its maximum during a water-rich period around September. It then enters a plateau after October until the subtidal period decreases to a minimum around April of

the following year [3–6]. Studies show that the QTP is one of the most sensitive regions to global climate change. The lakes, located in endorheic basins, are less affected by human activities and are outposts of the cryosphere and climate change [7,8]. Accurately depicting the lake area of the whole QTP is one of the concerns of the Intergovernmental Panel on Climate Change (IPCC, https://www.ipcc.ch/srocc/, accessed on 20 February 2022), which helps to enhance the understanding of climate change under the background of global warming through the temporal and spatial changes of lake waterbodies in the plateau. It provides scientific support for protecting and developing ecological resources in the QTP lake area and adopting climate change countermeasures. However, the traditional method of extracting the lake area by directly measuring water level and shoreline has a high cost, poor timeliness, and spatial accessibility. Due to the way monitoring data are acquired and stored varies from site to site, it is challenging to meet the requirements of current global change research on the integrity of hydrological monitoring data [9,10]. Because of its advantages of comprehensive spatial coverage, strong cyclicality, and low cost, remote sensing became an important means of lake waterbody area extraction. It has also extensively promoted the dynamic monitoring of lake areas on the QTP.

From the data perspective, satellite remote sensing provides many multi-source data with long-term series and high-spatial-resolution. The data sources commonly used in the research mainly include MODIS [11,12], Landsat [13–15], SPOT [16,17], ALOSE [18], ASTER [19,20], HJ-1A/1B [21,22], WorldView [23,24], QuickBird [25], IKONOS [26], GaoFen [27,28], ZY-3 [29,30], SAR [31], and hyperspectral data [32,33]. To better monitor global climate change, studies usually require an image covering the entire QTP at least once a year, with an image resolution preferably no greater than 32 m (capable of identifying lakes with an area of 1 km$^2$). In contrast, the image acquisition time is best in October, when the lakes are more stable. The Landsat series provided by the USG (USGS, http://glovis.usgs.gov/, accessed on 5 April 2022), due to its relatively high spatial resolution, long time series, free access, and rich data, became the first choice for lake area extraction of the QTP [34]. Landsat-9 data, accessible by the USGS on 10 February 2022, are identical in band setting to the Landsat 8 sensors, and include higher radiometric resolution (14-bit quantization increased from 12 bits for Landsat-8). Since Landsat-9 is matched with Landsat-8 in track design, if the two data are combined for analysis, the temporal resolution could be improved from 16 days to 8 days [35]. For large-scale research, the acquisition of images with good consistency and target recognition ability in the time window is pretty important. Landsat-9 data and Landsat-8 data, which are proved to be able to effectively identify waterbodies, have good complementarity in the revisit cycle and consistency in parameter settings. Therefore, Landsat-9 has great application potential in the field of large-scale lake mapping. However, at present, there is no comprehensive research on water area extraction in QTP based on Landsat-9 data. The research on water extraction based on Landsat-9 data is of great significance to the protection and development of plateau ecological resources and the research on global climate change.

Many scholars proposed various lake water feature extraction models for different application scenarios with multiple data, mainly divided into the single-band threshold method, spectral relation method, water index method, machine learning, and multiple models assisted synthesis method [36]. The single-band threshold method mainly uses the difference in reflectance between waterbodies and other ground objects in the infrared band. It selects water features with thresholds, and is widely used when early remote sensing data are scarce. For example, Frazier et al. [37] used the threshold method to extract the waterbody of Wagga Lake with the TM4, TM5, and TM7 bands, and the results show that the extraction method with TM5 band was better. The spectral relation method mainly extracts the waterbodies by analyzing the spectral curves of the characteristic band on multiple ground objects and constructing logical classification rules. For example, Du et al. [38] and Yang et al. [39] extracted the waterbody information by building a logical relation based on the characteristics of the gray value of the waterbody and the difference in reflectivity reduction between the waterbody and shadow in the blue-green

band, respectively. The spectral relation method can better distinguish between waterbodies and mountain shadows, but the anti-noise interference ability is poor, and it is susceptible to the influence of non-water noise, such as vegetation and buildings, which is the same as the single-band threshold method [4,40].

The water index method uses the bands with water reflection differences to construct the ratio operation, which suppresses the vegetation information, weakens the influence of soil, buildings, and shadows, and highlights the water information. Since Mcfeeters [41] proposed the normalized difference water index (NDWI), many scholars proposed a variety of improved water indexes according to different background features and water characteristics, such as the modified normalized difference water index (MNDWI) [42], enhanced water index (EWI) [43], automated water extraction index (AWEI) [14], shadow water index (SWI) [44], and land surface water index (LSWI) [45]. They drove the rapid development of water information extraction research [46–48]. Although the water index methods are simple to operate and can eliminate shadows cast from mountains, buildings, and vegetation, recognition errors of small waterbodies and the boundaries between water and land are large. The machine learning methods regard waterbodies as a category, using specific classification rules to classify them to obtain water features. Machine learning methods, such as minimum distance (MD) [49], decision tree (DT) [50], support vector machine (SVM) [51], neural network (NN) [52], random forest (RF) [53], and deep learning (DL) [54], are widely used. Machine learning algorithms realize the effective use of the spatial and texture information of high-spatial-resolution images and have a good effect on the extraction of small waterbodies, but the process is relatively complex. For example, Sui et al. [55] integrated three modules of initial extraction of water, iterative segmentation, and change detection with the help of GIS technology to realize the automation of the water extraction process; and Qiao et al. [56] proposed an adaptive extraction method of "whole-local" progressive spatial scale transformation based on NDWI and combined with the spectral feature fitting method and iterative algorithm to accurately extract the lake range. The multiple model-assisted synthesis method is used to synthesize a variety of models and methods to solve the problem of water extraction in large-scale and complex background scenes, but the process is complex, and the generalization ability of the model is poor. By expanding and analyzing the principles, advantages, and disadvantages of different water extraction methods, it is not difficult to see that the water extraction algorithm has no absolute advantages and disadvantages, and the consistency with the data source and applications will also affect the accuracy of the algorithm. The computational volume of large-range water extraction is quite large, so most previous studies focused on the threshold extraction algorithms with lower complexity, while relatively few studies focused on the machine learning algorithms with higher complexity. The previous study shows that the traditional NDWI method performed the best among the threshold extraction algorithms in water extraction of the QTP [6], but the relevant conclusions are not clear for Landsat-9 imagery, and the study did not include machine learning algorithms. Therefore, it is of certain practical significance to carry out the comparative study of various types of algorithms for plateau lake area extraction based on Landsat-9 images, which can provide a useful reference for the follow-up related research.

To sum up, the research on the remote sensing extraction of spatial distribution information of lakes in the QTP has crucial scientific significance for coping with global climate change. Although water extraction algorithms based on different resolutions and types of remote sensing data combined with other principles are widely used in water information extraction, there is not a comparative study of Plateau Lake area extraction algorithms based on Landsat-9 images. Based on cloud technology, Google Earth Engine (https://developers.google.cn/earth-engine, accessed on 1 April 2022) stores all kinds of data, with a total amount of more than 5 Pt on the cloud, and allows users to call the computing platform composed of tens of millions of computers through the web client to visually retrieve, process, analyze, and download all kinds of data online. Compared with the traditional data processing method based on local computers, the emergence of

GEE dramatically improved the efficiency of big data processing research and reduced the threshold of large-scale analysis [57–59]. Therefore, this paper uses GEE to conduct a comparative study of Landsat-9 lake area extraction in combination with ten widely used water extraction algorithms (including a single-band threshold extraction algorithm, two spectral correlation algorithms, four water index algorithms, and two machine learning algorithms). Through comparative research, this study quantitatively evaluates the consistency between different algorithms and Landsat-9 data in the plateau lake extraction scene to determine the algorithm for large-scale plateau lake area extraction suitable for Landsat-9, and provides some references and suggestions for further research in subsequent related fields.

This paper is structured as follows: Section 2 describes the study areas and the data; Section 3 provides a detailed and complete description of the experimental procedures and methods; Section 4 presents and analyzes the waterbody extraction results; Section 5 highlights the main findings and the implications of this study, and is followed by our conclusion.

## 2. Materials

### 2.1. Study Region

The QTP region, as shown in Figure 1, is our study area, which locates between 67°40′37″E~104°40′57″E and 25°59′30″N~40°1′0″N. The region's total area is over 3 million km$^2$, with an average altitude of about 4320 m, and it spans nine countries, including China, India, Pakistan, Tajikistan, Afghanistan, Nepal, Liberia, Myanmar, and Kyrgyz. [60]. This region is an important water resource reserve area in China. The annual outbound water volume of the rivers in Southwest China, which mainly originate here, accounts for about 95% of China's total annual water consumption (2020). In addition, there are many lakes in the region, with a total area of about 50,000 km$^2$, accounting for more than half of the lake area in China [3,5].

### 2.2. Dataset

The Landsat-9 satellite was successfully launched on 27 September 2021, and Landsat 9 data were publicly available on 10 February 2022. The Landsat-9 satellite carries the operational land imager (OLI) and the thermal infrared sensor (TIRS). The radiometric resolution of the sensor is improved to 14-bit quantization. Landsat-9 orbit has a time interval with the Landsat-8 and Sentinel-2 orbit, so multi-source data analysis can be carried out to improve time resolution [35]. This study uses the Landsat 9 Collection 2 surface reflectance (L9C2_SR), which is geometrically and radiometrically corrected by USGS and downloaded on the GEE platform. L9C2_SR provides data for eight spectral bands with a ground sampling distance (GSD) of 30 m, includes ultra-blue (0.435–0.451 μm), blue (0.452–0.512 μm), green (0.533–0.590 μm), red (0.636–0.673 μm), near-infrared (0.851–0.879 μm), shortwave infrared 1 (1.566–1.651 μm), shortwave infrared 2 (2.107–2.294 μm), and surface temperature (10.60–11.19 μm) bands. Landsat-9 was launched on 27 September 2021, and the available data could not overlap with the best observation period (October). Considering the influence of lake ice, the 1211 images used in this study are mainly from March and April, and a few from January and February. The coverage of images is shown in Figure 1c. The whole study area has image coverage, with a minimum of 6 times and a maximum of 45 times.

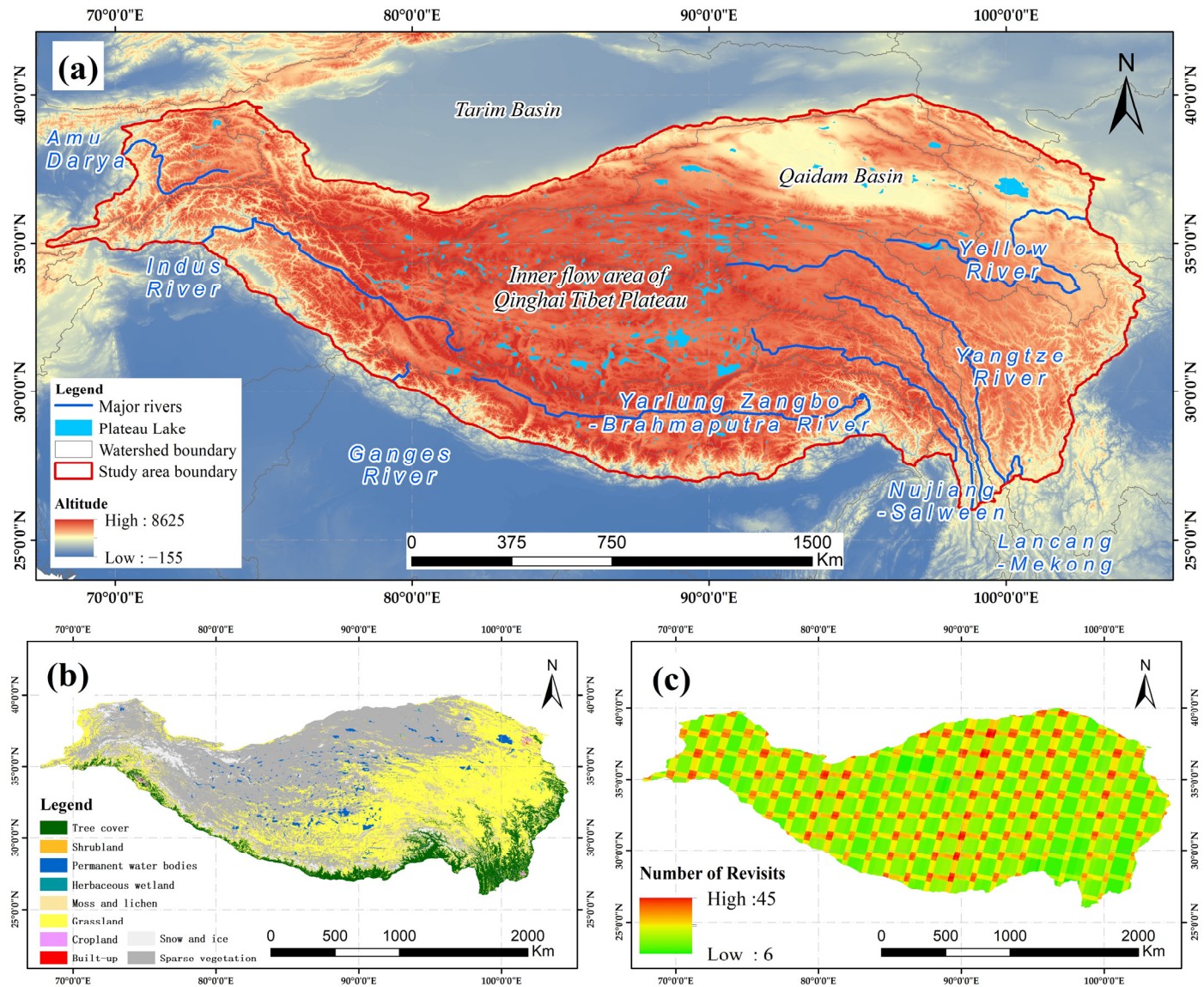

**Figure 1.** Study region and dataset. (**a**) Locations of the study area and the terrain characterized. Study area boundary data were mainly based on TPBoundary_2021. The primary river data were collected from HydroRIVERS. The lake data were based on TPLA_V3. The topographic data were produced by NASA-provided SRTM DEM [61]; (**b**) the landcover data source was ESA_WorldCover; (**c**) the coverage of Landsat-9 images in the study area was based on L9C2_SR.

The Qinghai–Tibet Plateau Lake area data set (V3.0) (TPLA_V3, http://data.tpdc.ac.cn, accessed on 15 February 2022) [62,63] released by the Institute of Tibetan Plateau Research was used to construct lake samples as reference truth values. TPLA_V3 contains a total of 15 time series of lake boundaries data of the QTP with an area of more than 1 km². These boundaries were delineated from Landsat MSS/TM/ETM+/OLI data for the 1970s (1972–1976, but mainly 1976), ~1990, ~1995, ~2000, ~2005, ~2010, and 2013–2021. The lake boundaries in the 1970s, ~1990, ~2000, and ~2010 were divided wholly based on visual interpretation of remote sensing images [63]. The other lake data sets were delineated by using the NDVI with an appropriate threshold. Visual checking against the original Landsat images and manual editing of incorrect lake boundaries were also employed [62]. Affected by runoff and precipitation, lakes on QTP will have seasonal area changes. According to the differences in lake types and area sizes, seasonal area changes are also considerable, with the most significant area change of more than 80 km² [6]. To reduce the uncertainty caused by seasonal changes, TPLA_V3 mainly uses the images of October in the high-water

season to extract the boundaries of lakes. Before 2013, the available images in October were limited, and a small amount of data in September and November were inevitably used. After 2013, the availability of the high-quality Landsat 8 data enabled annual lake mapping to be achieved. In this study, the lake boundaries in 2021 in TPLA_V3 were used as the parameter data. However, the data used for extracting the lake area in this study are mainly concentrated in March and April (a small part of the data in January and February are used for supplement) in the low-water season. To ensure that the reference truth data are sufficiently representative, the Landsat and Sentinel images of the same period are used for manual selection and adjustment when constructing the lake sample set for accuracy verification and the lake boundary sample set for area relative error analysis.

In addition, the following datasets were also used in this study: QTP boundary data (TP-Boundary_2021, http://www.geodoi.ac.cn, accessed on 15 February 2022) [60], WorldCover data with 10 m spatial resolution published by European Space Agency (European Space Agency WorldCover 10 m 2020 product, ESA_WorldCover, https://zenodo.org/record/5571936, accessed on 15 February 2022), global river water data released by the WWF Conservation Science Program and USGE in conjunction with several scientific institutions (vectorized line network of rivers, HydroRIVERS) [64]. Among them, TPBoundary_2021 data were used to determine the research regions; ESA_WorldCove was used to generate samples for machine learning algorithm classification; and HydroRIVERS combined ESA_WorldCover to construct non-lake waterbodies for post-classification processing.

## 3. Methods

The step-wise progression of waterbody area extraction is illustrated in Figure 2 and encompasses four steps: data collection and processing, classification, and evaluation.

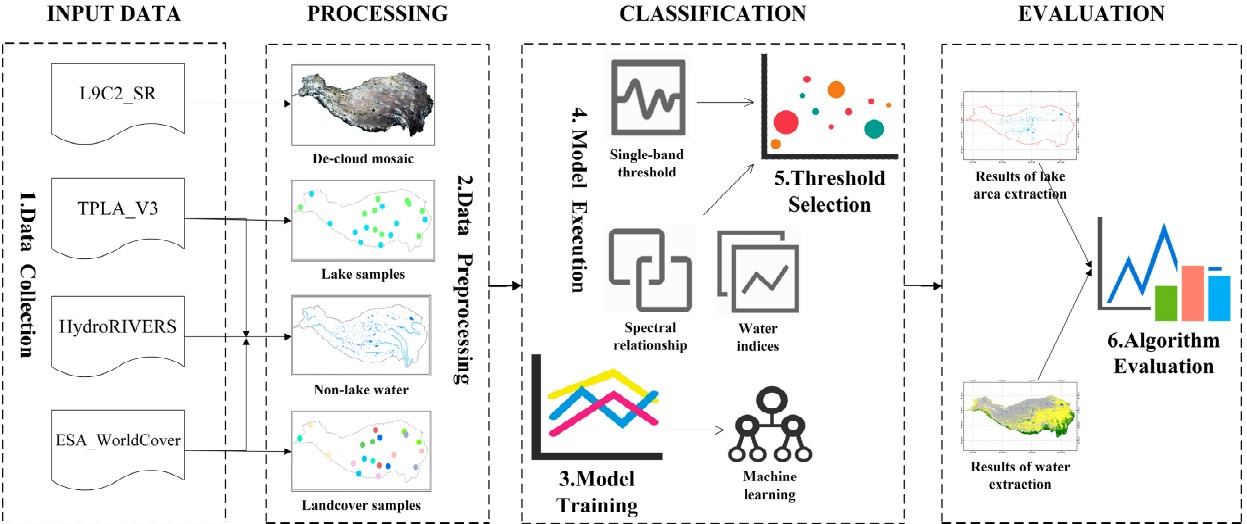

**Figure 2.** The general workflow of the experiment.

Step 1: Landsat-9 images, lake data, river data, and landcover data were collected as input data.

Step 2: The Landsat-9 images were processed by removing clouds and shadows and mosaicked into a de-cloud map. The remaining data were used to generate lake samples, landcover samples, and non-lake water regions, and finally formed the pre-processing data results.

Step 3: The model parameters and the thresholds were adjusted according to the lake and landcover samples. Based on specific model parameters and points combined with multiple models, waterbody extraction in the QTP study area was realized. Further, the waterbody extraction results were excluded by using non-lake waterbody data, and the lake waterbody extraction results in the study area were obtained.

Step 4: Finally, we evaluated the performance of algorithms with accuracy metrics based on the extraction information derived from step 3.

*3.1. Data Preprocessing*

Data preprocessing includes image de-cloud mosaic, classification sample construction, and non-lake water area extraction. To obtain high-quality cloud-free images covering the study area, cloud removal becomes a significant preprocessing step and significantly impacts waterbody extraction results. Landsat-9 images provided by GEE are organized in a data structure called "Imagecollection". Unlike traditional remote sensing images, which emphasize the data organization structure of the scene, "Imagecollection" emphasizes the concept of the real spatial location corresponding to the pixel. L9C2 contains all downloadable Landsat-9 images and forms a "stack"-like structure at the spatial position corresponding to the pixel. Users can use the functions provided by GEE to filter and sort the "stack" representing a specific spatial location based on time, pixel value, and other attributes. The spatial position of the cloud and the sun glint changes with time and is shown as an abnormally high value of pixels on the image. Therefore, we use the function to sort the values of all revisited images in the time range pixel by pixel, and select the median value as the pixel value of the point to build an image without the influence of cloud and sun glint. This study used the "CLOUD_COVER" attribute and the "QA_PIXEL" band included in the L9C2_SR to detect clouds and cloud shadows as much as possible and mask them [65], and then used functions such as median() and min() to mosaic the masked image in order to synthesize a cloud-free image. Due to L9C2_SR having limited images in the time interval from 1 January 2022 to 30 April 2022, the image cloud coverage in January and February is large. Therefore, directly setting the attributes of "CLOUD_COVER", "QA_PIXEL", or "IMAGE_QUALITY_OLI", simply using functions such as median() and mean() to mosaic cloud-free images, will lead to poor cloud shadow removal results or the mosaic image no being able to cover the study area. Through the analysis of dataset images and many comparative experiments, we found that by using "QA_PIXEL " to obtain the pixels affected by clouds and cloud shadows detected by the Cfmask algorithm [66] and masking them without the "CLOUD_COVER" attribute setting, and we can then obtain mosaic images with good cloud removal effect and complete coverage of the study area through the following piecewise mosaic steps: adopting the minimum value in the area with revisit times less than 12; taking the quartile in the area where the number of revisits is greater than or equal to 12 and less than 30; and taking the median value of the area with the number of revisits greater than or equal to 26 for image mosaic. The results are shown in Figure 3.

Referring to the algorithm proposed by Deng et al. [57], the ESA_WorldCover (global land cover dataset) and the TPLA_V3 QTP (lake dataset) were used for hierarchical automatic random sampling [58]. The details of sample construction are shown in Table 1. ESA_Worldcover divides the land cover into 11 categories. After removing the mangroves that do not exist in the Qinghai–Tibet Plateau, combined with TPLA_ V3 Lake data, stratified random samples were generated. According to Landsat-9 images of Google Earth cloud removal and Sentinel-2 images of the same period with higher spatial resolution, the generated random samples were manually revised to eliminate wrong samples. The samples were randomly divided into a test set, training set, and validation set according to the ratio of 3:4:3. The test set was used for the search and optimization of model hyperparameters, the training set was used for model training, and the validation set was used for model accuracy verification. In contrast, for non-machine learning models, the test set and training set were used to determine the threshold, and the validation set was used for accuracy evaluation.

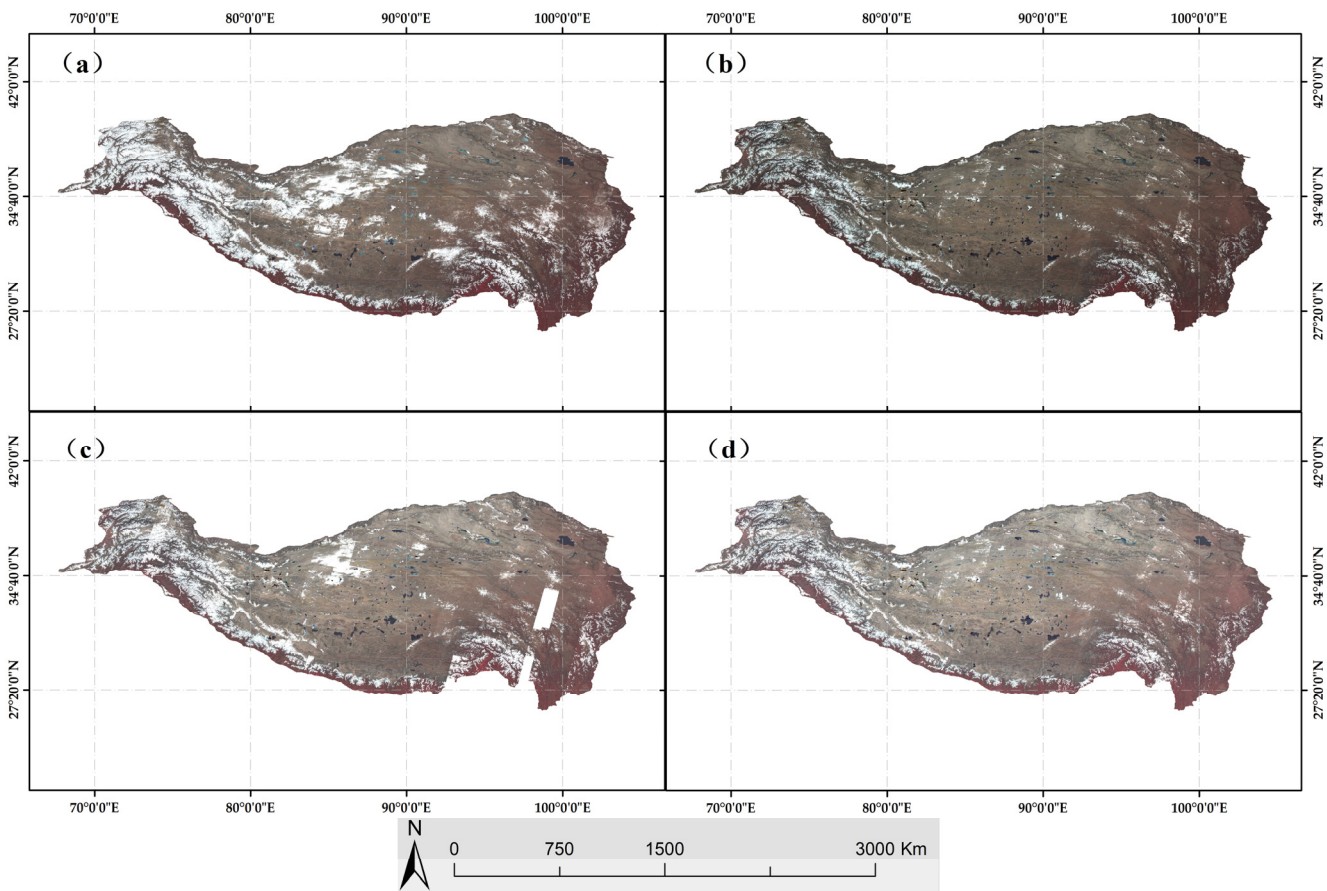

**Figure 3.** Mosaic images under removing clouds were synthesized in false color in bands 5, 4, and 3, and stretched with 2.5 standard deviations. (**a**) The cloud removal mosaic image with QA_PIXEL band, IMAGE_QUALITY_OLI wat set to 9, and median function. However, there are still a lot of clouds in the northwest and southeast regions. (**b**) The cloud removal mosaic image with QA_PIXEL band, IMAGE_QUALITY_OLI wat set to 9, and min function. The result shows that the cloud removal effect is better, but the shadow effect is enhanced. (**c**) The cloud removal mosaic image with QA_PIXEL band, CLOUD_COVER wat set to 20, and min function. Mosaic results cannot cover the study area, and some clouds are still obscuring the northwest. (**d**) The cloud removal mosaic image with QA_PIXEL band and piecewise mosaic. The results show that the cloud removal effect is relatively apparent, the shadow enhancement effect is relatively small, and the coverage of the study area is complete.

**Table 1.** Landcover samples.

| Landcover Class | Landcover ID | Number | Source |
|:---:|:---:|:---:|:---:|
| Tree cover | 1 | 11,127 | ESA_WorldCover |
| Shrubland | 2 | 4345 | ESA_WorldCover |
| Grassland | 3 | 13,210 | ESA_WorldCover |
| Cropland | 4 | 7376 | ESA_WorldCover |
| Build-up | 5 | 1767 | ESA_WorldCover |
| Bare/sparse vegetation | 6 | 15,972 | ESA_WorldCover |
| Snow and Ice | 7 | 8409 | ESA_WorldCover |
| Waterbodies | 8 | 41,654 | TPLA_V3 and ESA_WorldCover |
| Herbaceous | 9 | 2437 | ESA_WorldCover |
| Moss and lichen | 10 | 10,726 | ESA_WorldCover |

A previous study combined the reservoir, dam database, river database, and other non-lake waterbodies to extract the lake waterbody area based on the extraction results of waterbodies [67]. Therefore, this study combined ESA_WorldCover, TPLA_V3, and

HydroRIVERS data to establish a non-lake waterbody dataset for post-processing waterbody extraction. Non-lake permanent waterbodies were obtained by a geometric union of waterbodies between TPLA_V3 and ESA_WorldCover after erasing the intersection waterbodies. The buffer regions were built according to the inter-annual variations of the lake [6], and the buffer regions were used to erase the HydroRIVERS data to obtain the river dataset. Finally, we formed the non-lake waterbody dataset for post-processing.

### 3.2. Waterbody Extraction Algorithms

Waterbody extraction algorithms, such as the single-band threshold, spectral relation, water index, and machine learning are widely used, especially the water index method and machine learning [50]. Based on previous research [9,68], we selected the most commonly used four water index models, two machine learning models, two spectral relationship models, and a single-band threshold model as the target algorithm to explore the extraction results of the waterbody and lake waterbody in the QTP based on Landsat-9 data and to determine the best performance of waterbody area extraction algorithms under the background of the large-scale study region. The detailed information of each model is shown in Table 2. Where $\rho_i$ represents the band $i$, which refers to Section 2.2. $N_1$ and $N_2$ are the experience thresholds. $X$, $C$, $gamma$, $nt$, and $mf$ represent the feature sets (including 8 bands of Landsat-9 image and various water indexes.), regularization parameter (known also as penalty factor), kernel width, the number of decision trees, and the number of input features used to split the nodes, respectively.

**Table 2.** Details of waterbody extraction algorithms.

| Model Type | Name | Formulas/Parameters | References |
|---|---|---|---|
| Single-band threshold method | SBT | $\rho_{SWIR1} < N_1$ | Frazier and Page [37] |
| Spectral relationship method | SR | $\rho_{\text{green}} + \rho_{red} > \rho_{NIR} + \rho_{SWIR1}$ | Du and Zhou [38] |
| | mSR | $(\rho_{\text{green}} + \rho_{red}) - (\rho_{NIR} + \rho_{SWIR1}) - (\rho_{\text{blue}} - \rho_{green}) > N_2$ | Yang, et al. [39] |
| Water indices model | NDWI | $NDWI = (\rho_{\text{green}} - \rho_{NIR})/(\rho_{\text{green}} + \rho_{NIR})$ | Mcfeeters [39] |
| | mNDWI | $mNDWI = (\rho_{\text{green}} - \rho_{SWIR1})/(\rho_{\text{green}} + \rho_{SWIR1})$ | Xu [41] |
| | AWEIns/AWEIs | $AWEI_{ns} =$ $4 \times (\rho_{\text{green}} - \rho_{SWIR1}) - (0.25 \times \rho_{NIR} + 2.75 \times \rho_{SWIR2})$ $AWEI_s =$ $\rho_{blue} + 2.5 \times \rho_{\text{green}} - 1.5 \times (\rho_{NIR} + \rho_{SWIR1}) - 0.25 \times \rho_{SWIR2}$ | Feyisa, et al. [14] |
| | mAWEI | $mAWEI = 5 \times (\rho_{\text{green}} - \rho_{NIR}) + (\rho_{blue} + \rho_{red} - 4 \times \rho_{SWIR2})$ | Nie, et al. [67] |
| Machine learning model | SVM | $X, C, gamma$ | Razaque, et al. [51] |
| | RF | $X, nt, mf$ | Ko, et al. [53] |

### 3.3. Model Parameters and Thresholds

Previous studies show that the classification accuracy of the machine learning models is mainly dependent on model hyper-parameters [69]. To effectively adjust the hyper-parameters and optimize the model's classification accuracy, drawing on the ideas proposed by Porwal et al. [70], this study used the sklearn package, the open-source machine learning toolkit. First, the RandomizedSearchCV searched for an optimal solution in the large-scale range and then used GridSearchCV to search for a certain floating fine-tuning of the hyper-parameters within the small-scale range. Further, the 5-fold cross-validation method was used to verify the classification performance of the test set on the model, and the average accuracy was regarded as the estimated value for fine-tuning the parameter optimization.

In Table 2 of Section 3.2, the SVM model uses radial basis function (RBF) kernel and needs to search and optimize the parameters of $C$ and $gamma$. The random forests (RF) model uses classification and regression trees (CART) as the basic algorithm, and $nt$ $md$ parameters must be adjusted and optimized.

Appropriate thresholds are critical for water extraction models based on threshold segmentation. The selection of points has a certain randomness, which varies with the type of index, the identification scenario, and the subpixel water/non-water components [71]. The OTSU method [72] and its improved algorithms [73–75] are commonly used for water index threshold automatic extraction. They use waterbodies and near-water land to present a bimodal distribution in the frequency domain of the index to set the optimal threshold in order to split the image into "foreground" and "background" to achieve the classification of waterbodies and non-waterbodies [76]. However, for the extraction of waterbodies and lake waterbodies in the QTP region, there are two problems with this threshold algorithm. First, there is a huge difference in the proportion of waterbodies and non-waterbodies in the region, which makes it impossible to effectively classify waterbodies in the frequency domain when they are covered by other landcover classes [73]. Second, the area of the study region exceeds 3 million km$^2$, and using such threshold segmentation methods requires high computing power. Therefore, this study uses lake samples to adjust the threshold manually.

### 3.4. Evaluation Metrics

In this study, the extraction accuracy of water and lakes is verified by the confusion matrix with a verification set, and the performance of the above water extraction model is quantitatively evaluated through four evaluation metrics, including overall accuracy, kappa, map-level accuracy, and user's accuracy. Among them, the overall accuracy reflects the overall effect of the algorithm. Kappa indicates the consistency between the ground truth data and the predicted value. Map-level accuracy represents the probability that the validation sample is correctly classified. Finally, the user's accuracy means the ratio of the inspection points falling on Category *i* on the classification diagram to be correctly classified as Category *i* [74]. In addition, the error analysis formula is introduced to evaluate the accuracy of area extraction.

$$\delta = \frac{|A_r - A_e|}{A_r} \tag{1}$$

where $\delta$ is the error result. $A_r$ represents the ground truth area and $A_e$ is the area extracted by the algorithm.

## 4. Results and Discussion

### 4.1. Parameters and Thresholds Selection Results

#### 4.1.1. Optimization Parameters in Machine Learning Methods

For the machine learning methods, we evaluated the importance of the sample features involved in classification. We reduced the redundancy between sample features to improve the efficiency and accuracy of the algorithms. The Gini index, also known as Gini impure, is an indicator to measure the probability of random samples being misclassified in the sample set. The smaller the Gini index, the smaller the probability that the selected samples in the set will be misclassified. When all samples in the set are of one class, the Gini index is 0. Each tree in the random forest algorithm is a CART decision tree. When the tree chooses to use a feature to split down, it needs to calculate the Gini index to measure the purity of the sample set before and after the split. The smaller the Gini index of the left and right branches after splitting, the higher the accuracy of separating using this feature. Suppose that node m on a tree uses a feature to split down, the Gini index before splitting is $GI$, and the Gini index of the left and right branches after splitting is $GIL$ and $GIR$, respectively. On this decision tree, this feature is split $k$ times, and $n$ trees in the whole forest use this feature, so the importance of this feature in the entire forest is $\sum_{j=1}^{n} \sum_{i=1}^{k} [GI - (GIL + GIR)]_m$. The importance score of this feature is obtained by dividing the calculated result of this feature's significance and the sum of the importance of all features. In the sample construction stage, we used the test dataset constructed in Section 3.1 to construct the waterbody and non-waterbody samples. We referred to [77] to evaluate the importance of eight bands and seven

water indexes that participated in the calculation based on the Gini index. According to the importance score, we successfully put the features into the classifier for cross-accuracy verification to determine the optimal feature parameters for water extraction.

The evaluation and selection results of characteristic parameters are shown in Figure 4. Among the features represented by the horizontal axis of Figure 4, the importance score of features gradually decreases from left to right. The highest and lowest important score features are SR_B5 and AEWIns, respectively. The low importance scores of AEWIs, AEWIns, and SR_B6 may be due to these features entering the classifier late and having a significant correlation with the previous parts, which leads to the information being judged as useless information [77]. Further, according to the overall accuracy of cross-validation, it can be seen that when the total number of features reaches six, the overall accuracy reaches 93.80%, and the remaining features make a less cumulative contribution to the improvement of accuracy. Based on the importance scores and overall accuracy results, the feature sets for water extraction methods (RF and SVM classifiers) are defined as {SR_B5, NDWI, SR_B7, mNDWI, mAEWI, and SR_B6}.

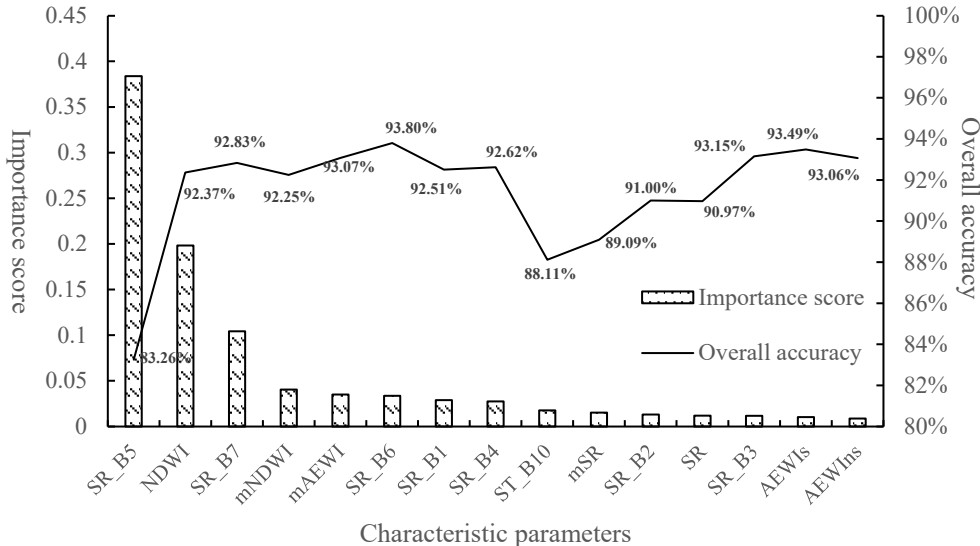

**Figure 4.** Evaluation and selection results of characteristic parameters.

To improve the accuracy of machine learning models, we optimized the critical parameters of SVM and RF algorithms based on the above-constructed feature sets. The key parameters of the SVM algorithm include the *C* regularization parameter, kernel function types, and their related kernel parameters. Standard kernels with SVM classifiers include the polynomial kernel, radial basis function (RBF) kernel, linear kernel, and more. Previous studies show that the RBF kernel has better performance for image recognition with prominent non-linear characteristics [78]. Therefore, we selected the RBF kernel as the kernel function and optimized the parameters of *C* and *gamma*. *C* is the relaxation vector parameter in the SVM classifier. When the *C* value is small, the interface is smooth.

On the contrary, when it becomes large, the complexity of the model will increase. The kernel parameter *gamma* defines the magnitude of the effect of a training sample, which is the reciprocal of the width of the RBF kernel. When the *gamma* value is more significant, the influence on the radius is more minor, and overfitting is easy. On the contrary, underfitting easily occurs [79]. The effective range of the *C* value and *gamma* value is $10^{-8} \sim 10^8$, but in practical application, the possible optimal values are generally in the range of (0.1, 100) and (0.0001, 10) [80]. Therefore, we used the RandomizedSearchCV function to search for the optimal parameters *C* and *gamma* values in the interval of (0.1, 100) and (0.0001, 10) with the multiple of 10 as the step size and determined that the optimal values appear around 10~100 and 0.01~0.1, respectively. Then, we used the GridSearchCV function to realize grid search in steps 10 and 0.01 between (10, 100) and (0.01, 0.1). Figure 5a shows that the

overall accuracy reaches the maximum value of 95.79% when $C = 40$ and $gamma = 0.04$. Therefore, the experiment will train the SVM classifier with these parameters.

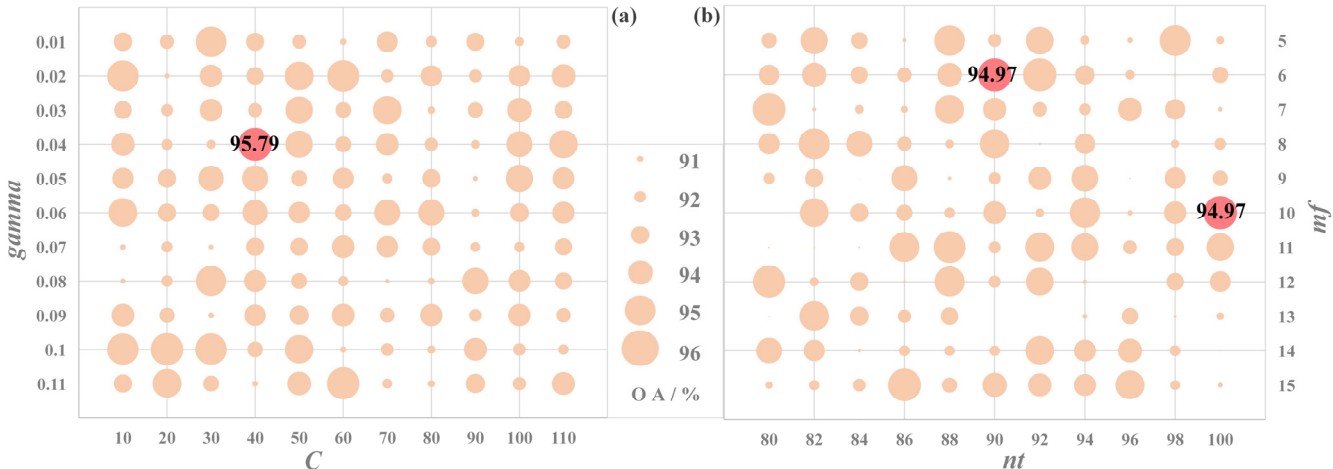

**Figure 5.** Optimization of characteristic parameters based on grid search. (**a**) Optimal parameters retrieval graph for SVM; (**b**) optimal parameters retrieval graph for RF.

The random forest (RF) classifier is widely used as an ensemble learning classifier in remote sensing information extraction. Randomness in the RF model is mainly reflected in the random selection of datasets and features used in each tree. The corresponding parameters are the number of decision trees ($nt$) and the maximum number of features ($mf$) to be selected for the node split when growing the trees [81]. When $nt$ is larger, more decision trees are involved, and the algorithm is more complex. The $mf$ parameter allows each tree to be trained to use only some features at random, reducing the overall operation and allowing each tree to focus on its chosen features. The research shows that a larger $nt$ and $mf$ will reduce the randomness and operation efficiency of the RF model and contribute less to the improvement of accuracy. Generally, the values of $nt$ and $mf$ will not exceed 1000 and 50, respectively [81,82]. As mentioned above, two parameters need to be set to produce the forest trees in this study. According to optimization methods of the parameters in SVM, we determine that the optimal values of $nt$ and $mf$ appear from 80 to 100 and from 5 to 15, respectively. We performed a grid search on the parameters in steps 2 and 1, respectively. Figure 5b shows that the overall accuracy reaches the maximum value of 94.97% when $nt = 90$ and $mf = 6$, or when $nt = 100$ and $mf = 10$. Considering that a larger $nt$ and $mf$ will increase the complexity of the RF model and reduce the generalization ability of the model, $nt = 90$ and $mf = 6$ are selected as RF parameters.

### 4.1.2. Selection of Waterbody Extraction Thresholds

We extracted the corresponding waterbody index information for the samples of the test set and training set constructed in Section 3. We combined the sample labels and waterbody index information to generate a confusion matrix, quantitatively verify the accuracy of waterbody extraction with different thresholds, and explore the best segmentation threshold for other waterbody indexes to improve the accuracy of waterbody extraction and reduce the subjectivity and contingency of the artificial threshold. To divide the threshold, we rewrite the formula of the two spectral relationship algorithms into the equation form: $SR = (\rho_{\text{green}} + \rho_{red}) - (\rho_{NIR} + \rho_{SWIR1})$ and $mSR = (\rho_{\text{green}} + \rho_{red}) - (\rho_{NIR} + \rho_{SWIR1}) - (\rho_{\text{blue}} - \rho_{green})$. Among the eight waterbody extraction models based on threshold segmentation, except for the single band threshold (SBT) extraction algorithm, which defines the area less than the threshold as waterbodies, the other models represent the area greater than the threshold as waterbodies. The variation in the waterbody extraction accuracy of each algorithm model with the thresholds is shown in Figure 6.

The overall accuracy of the water body extracted by the threshold

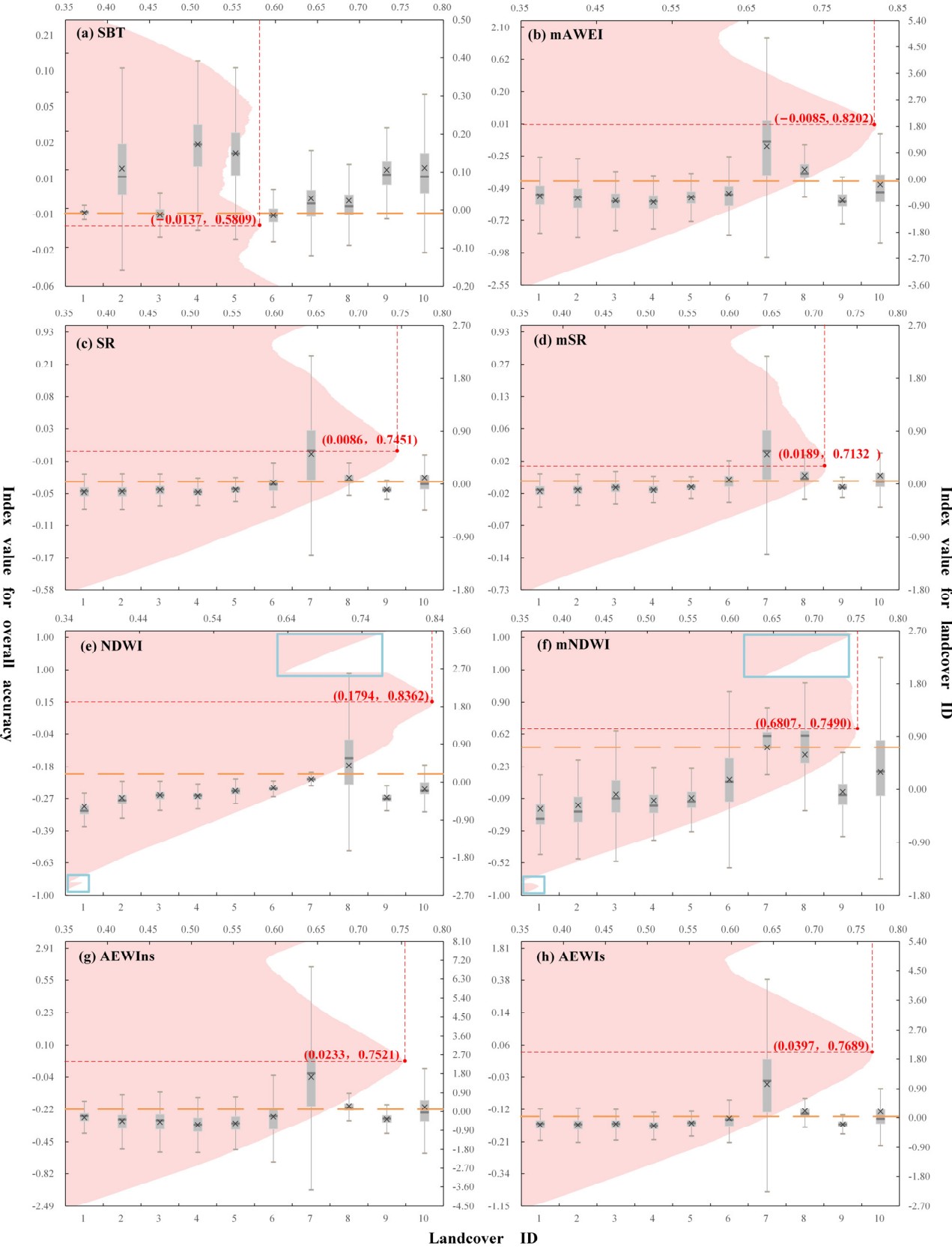

**Figure 6.** The threshold segmentation extracts the overall accuracy of the waterbody. The bottom X-axis represents the landcover code (as shown in Table 1), corresponding to the Y-axis on the right and the box

diagram in the figure, and the Y-axis on the right represents the value of different land classes in the water index calculated according to the formula in Table 2. In these box diagrams, boxes are interquartile ranges. Whiskers indicate the 1.5 times interquartile range. The horizontal line and cross sign in the boxes represent the median and mean, respectively. The top X-axis is matched with the left Y-axis to reflect the overall accuracy of segmentation results using different thresholds. The red dotted line intersection coordinates represent the best threshold and overall accuracy of the model, and the yellow dotted line represents the position of the best segmentation threshold on the box diagram. Each subfigure (**a**–**h**) represents the threshold selection process of the algorithm identified in its upper left.

In the overall accuracy of waterbody extraction results based on the threshold segmentation models, it can be found that except for the SBT model, the best overall accuracy of other threshold segmentation models for water extraction reached more than 70%. The NDWI threshold segmentation model has the highest overall accuracy for waterbody extraction at 83.62%, followed by mAEWI (82.02), AEWIs (76.89%), AEWIns (75.21%), mNDWI (74.90%), SR(74.51), and mSR (71.32%). The ranking of the overall accuracy of waterbody extraction results is highly consistent with the conclusions obtained in the previous study [6]. The SBT model relies on the low reflection of water at the 1.5~1.6 μm spectrum to extract waterbody, but the 1.5~1.6 μm spectrum is also at the absorption peak of vegetation [38]. As shown in Figure 6a, it can be seen that the threshold selection of −0.0137 cannot effectively distinguish water from landcover class 1 (tree) and class 3 (grassland), so its overall accuracy is only 58.09%. In addition, when the threshold value is −0.06, the overall accuracy of the SBT extraction result reaches its highest, at 63.77%. However, there are many true negative (TN) samples in this waterbody extraction result, and all samples are classified as non-waterbodies, so the threshold segmentation results fail to reach the requirements of waterbody extraction.

Additionally, the optimal segmentation thresholds selected by the overall accuracy can effectively distinguish waterbodies from other landcover classes except for snow and ice, which can be mutually confirmed with some conclusions of reaches [6,83]. At the same time, it also shows that the threshold selection method, based on the overall accuracy, has high accuracy. By analyzing the selection of threshold values of each model, it can be seen that the thresholds of mAEWI, SR, mSR, AEWIs, and AEWIns models are close to zero, which are −0.0085, 0.0086, 0.0189, 0.0397, and 0.0233, respectively, with strong anti-interference, consistent with the principle of each algorithm design [14,38,39,67].

Finally, comparing the blue boxes in Figure 6e,f, it can be seen that when the thresholds of NDWI and mNDWI models are close to −1 and 1, the overall accuracy is significantly distorted. This is because there are many abnormal values higher than one and some lower than −1 in the waterbody samples when using the surface reflection data for the division band operation. During the abnormal value processing, the values more significant than 1 and less than −1 are reclassified as 1 and −1, resulting in the accumulation of many waterbody samples at the thresholds of 1 and −1. A large number of outliers may be due to the influence of lake ice and the use of the min function in major areas during band cloud removal synthesis, which expands the shadow effect and increases abnormal low values.

### 4.2. Analysis of Waterbody Extraction Results

Based on the above optimization results of the machine learning model parameters, we used the training samples to train and classify the SVM classifier and RF model, reclassified the results into waterbodies and non-waterbodies, and calculated the confusion matrix on the validation set to evaluate the accuracy of waterbody extraction. Based on the optimal thresholds determined using the test set in 4.1.2, the whole study area was subjected to threshold segmented water extraction, and the extraction results of each algorithm were validated with accuracy using the validation set.

The accuracy validation results and the running time for various algorithms are shown in Table 3. By analyzing the overall accuracy of different algorithms, it can be seen that the accuracy validation results are highly consistent with parameter optimization and threshold selection results. On the one hand, RF and SVM algorithms have the highest classification accuracy, achieving an overall accuracy of over 95%. This is significantly better than other threshold segmentation waterbody extraction algorithms, which are essentially linear in the spectral domain. At the same time, the RF and SVM with RBF kernel can realize the non-linear segmentation extraction in the spectral domain by using ensemble learning and non-linear kernels, which have more robust adaptability and generalization to the non-linear spectral domain features of different landcover classes. On the other hand, the order of the overall accuracy of each threshold segmentation waterbody extraction algorithm is consistent with the threshold selection results. Generally, it has an improvement of about 5%, and the overall accuracy of SBT model increased by about 10%. This is because in the threshold segmentation waterbody extraction algorithms, the landcover classes of ice and snow, moss and lichen, tree cover (as shown in Figure 6), which are greatly confused with the waterbody threshold, account for a large proportion in the samples. Therefore, we selected 70% of the samples to calculate the overall accuracy when selecting thresholds. As a result, the sample imbalance is large, and the overall accuracy is low. The data set used for validation accounts for 30% of the samples, and the sample imbalance is reduced, so the overall accuracy is improved.

**Table 3.** Accuracy and running time of waterbody extraction with different models.

| Model | Cover Type | Producer's Accuracy (%) | User's Accuracy (%) | Overall Accuracy (%) | Kappa Coefficient | Running Time (min) |
|---|---|---|---|---|---|---|
| SBT | Non-water | 64.21 | 85.97 | 68.93 | 0.5331 | 22 ± 3 |
| | Water | 75.42 | 52.01 | | | |
| SR | Non-water | 74.75 | 89.99 | 76.26 | 0.4880 | 22 ± 3 |
| | Water | 79.91 | 56.69 | | | |
| mSR | Non-water | 70.36 | 91.34 | 74.31 | 0.4663 | 22 ± 3 |
| | Water | 83.87 | 53.92 | | | |
| NDWI | Non-water | 87.81 | 97.66 | 89.89 | 0.7720 | 22 ± 3 |
| | Water | 94.91 | 76.31 | | | |
| mNDWI | Non-water | 77.41 | 87.35 | 76.09 | 0.4654 | 22 ± 3 |
| | Water | 72.89 | 57.16 | | | |
| AWEIns | Non-water | 80.42 | 93.57 | 82.24 | 0.6099 | 22 ± 3 |
| | Water | 86.64 | 64.67 | | | |
| AWEIs | Non-water | 78.73 | 97.24 | 83.37 | 0.6461 | 22 ± 3 |
| | Water | 94.59 | 64.79 | | | |
| mAWEI | Non-water | 94.52 | 88.57 | 87.50 | 0.6828 | 22 ± 3 |
| | Water | 70.50 | 84.19 | | | |
| SVM | Non-water | 95.60 | 99.55 | 96.59 | 0.9198 | 154 ± 40 |
| | Water | 98.96 | 90.30 | | | |
| RF | Non-water | 95.21 | 98.87 | 95.84 | 0.9021 | 76 ± 15 |
| | Water | 97.36 | 89.38 | | | |

Secondly, analyzing the user's accuracy and producer's accuracy of different models, it can be seen that almost all models have relatively high producer accuracy and low user accuracy of waterbody extraction. In contrast, the user accuracy of non-waterbodies is high, and the producer accuracy is low. This shows that in large-scale regions of the QTP, the omission error of waterbody extraction is not high, and the commission error is mainly concentrated in the misclassification of a large number of non-waterbodies (ice, snow, vegetation, and shadows) into waterbodies. Comparing the kappa coefficients of different models, it can be seen that the prediction results of machine learning models are highly consistent with the actual results, indicating that the models are relatively stable. In the threshold segmentation waterbody extraction algorithms (except NDWI and mAEWI), the predicted results are less consistent with the actual results, indicating that the models are highly random and unstable.

Additionally, compared with other threshold segmentation waterbody extraction studies [6,14,37–39,41], it can be seen that the accuracy of threshold segmentation waterbody extraction models in this paper is relatively low. Comparing the experimental design and process of each study, we consider that there are two main reasons for this phenomenon. The first reason is the scale. Compared with other studies, the range of threshold segmentation waterbody extraction in this study exceeds 3 million km$^2$, is more extensive, and the background structure is more complex, so the accuracy of waterbody extraction is reduced. The second reason is the data pre-processing in this study. To reduce the cloud cover and improve the coverage of the Landsat-9 images in the study region, the method of segmented synthesis is used in this experiment by using the min function, which produces some low-value noise and reduces the data quality, thus affecting the accuracy of the threshold segmentation of waterbody extraction. It should be emphasized that the generation of low-value noise is balanced for various models. During waterbody extraction, the lake ice is in a state of melting, but not completely melting, making the transmission process of electromagnetic waves more complex and having a certain impact on the extraction of lakes. Although it will affect the accuracy of waterbody extraction, it does not affect the reliability of algorithm comparison results.

Finally, to better evaluate each algorithm's efficiency, we conducted four experiments on each algorithm and recorded its running time. Comparing the running time of different algorithms, it can be seen that the running time of the threshold extraction algorithm is the shortest, all of which are $22 \pm 3$ min. RF is also faster and can control the running time within one and a half hours. SVM has the lowest efficiency, and the longest running time is more than three hours. The threshold extraction algorithm can be divided into three steps: water index calculation, threshold determination, and water extraction. Water index calculation and extraction are completed on the GEE platform, which can be completed in 2 to 5 min according to the network speed and the GEE's computing power distribution. Threshold determination needs to import the sample into Excel and be calculated by a formula. The time is generally controlled at 19 min. Machine learning algorithms can also be divided into three steps: classification feature calculation, parameter adjustment, and water extraction. Similar to the threshold extraction algorithm, parameter adjustment must be calculated by importing samples into Python. The time required for RF and SVM is about 25 and 42 min, respectively. In comparison, the running time for classification feature calculation and water extraction is controlled at 36–66 min and 72–152 min, respectively, according to the network speed and computing power distribution. For large-scale research, such as lake extraction on the QTP, which has less strict requirements on running time, a machine learning algorithm with higher accuracy is a better choice.

To more intuitively show the effect and performance of each model for water extraction, this study selects Qinghai Lake Yamzho (the largest lake in the QTP), Yumco Lake (complex lake bay morphology), Yellow River (complex river morphology and background), and Brahmaputra River (complex river morphology and background) to compare the results of lake waterbody extraction. From left to right, the white oval areas on the imageries in Figure 7 are A1, A2, A3, a4, A5, A6, A7, A8, and A9. Dark areas are evident in large lakes with more extensive water surfaces, and the following three reasons most likely explain the occurrence of these areas: first, shadowing is enhanced by the min () function used in cloud-free image synthesis; second, the reflectance in the deep water zones of large lakes is intrinsically low; and third, lake ice within the lake is unfused. The occurrence of black areas is also one of the important reasons for the low extraction accuracy in waterbodies. Comparing the marine areas from A1 to A5 prone to omission error, it can be seen that SVM and RF algorithms can identify and extract these waterbody areas well.

In contrast, the threshold segmentation waterbody extraction algorithm has more omission errors. However, NDVI and AEWIs perform relatively better as threshold segmentation waterbody extraction algorithms. There are small waterbodies and prominent shadows in the A6 area. Except for the NDVI and AEWIs methods, all algorithms can better identify the shades in this area, but MSR and mNDWI models perform poorly in identifying small waterbodies. There are apparent floodplain wetlands in A7, A8, and their upstream areas due to the lateral movement of rivers. Except for the SVM, RF, NDWI, and AEWIns algorithms, other algorithms have commission errors to recognize wetlands as waterbodies. Finally, the A9 area is an abnormal area caused by the image synthesis algorithm. Different algorithms can better resist these abnormal values, except the mAEWI, mNDWI, MSR, and SBT models. In conclusion, we found that the SVM, RF, and NDWI models can better recognize waterbodies in different lake waterbody regions.

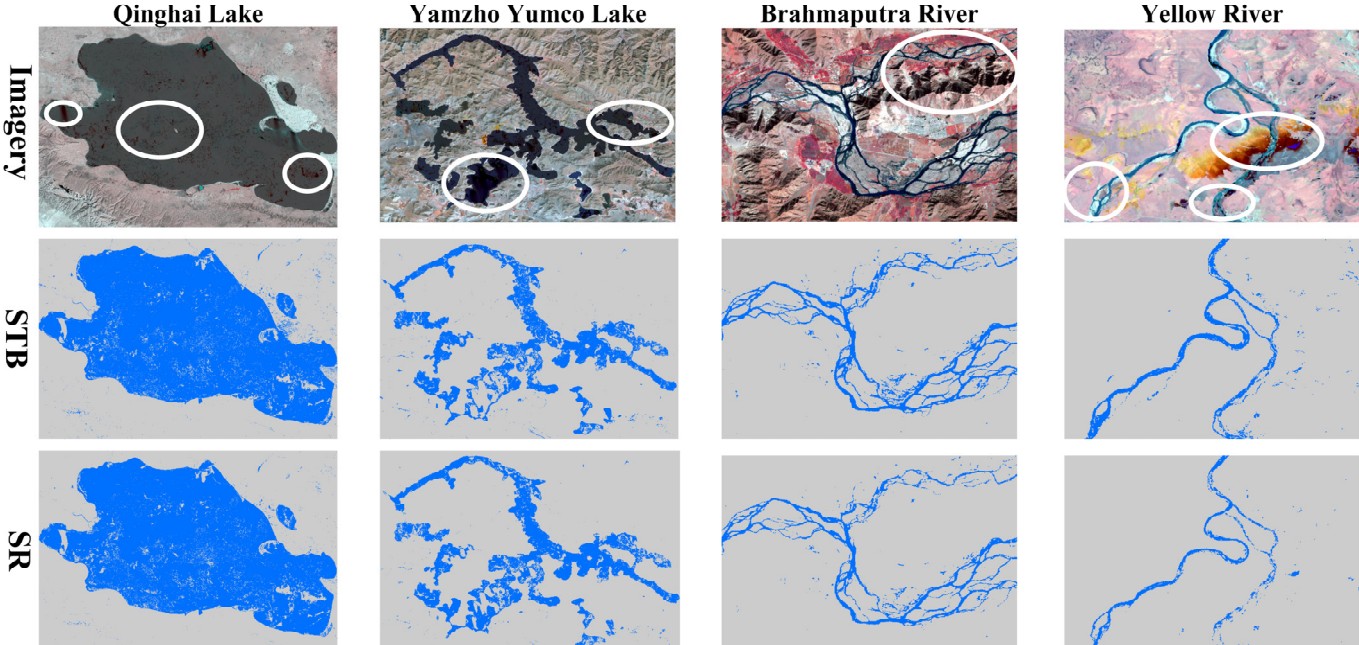

**Figure 7.** *Cont.*

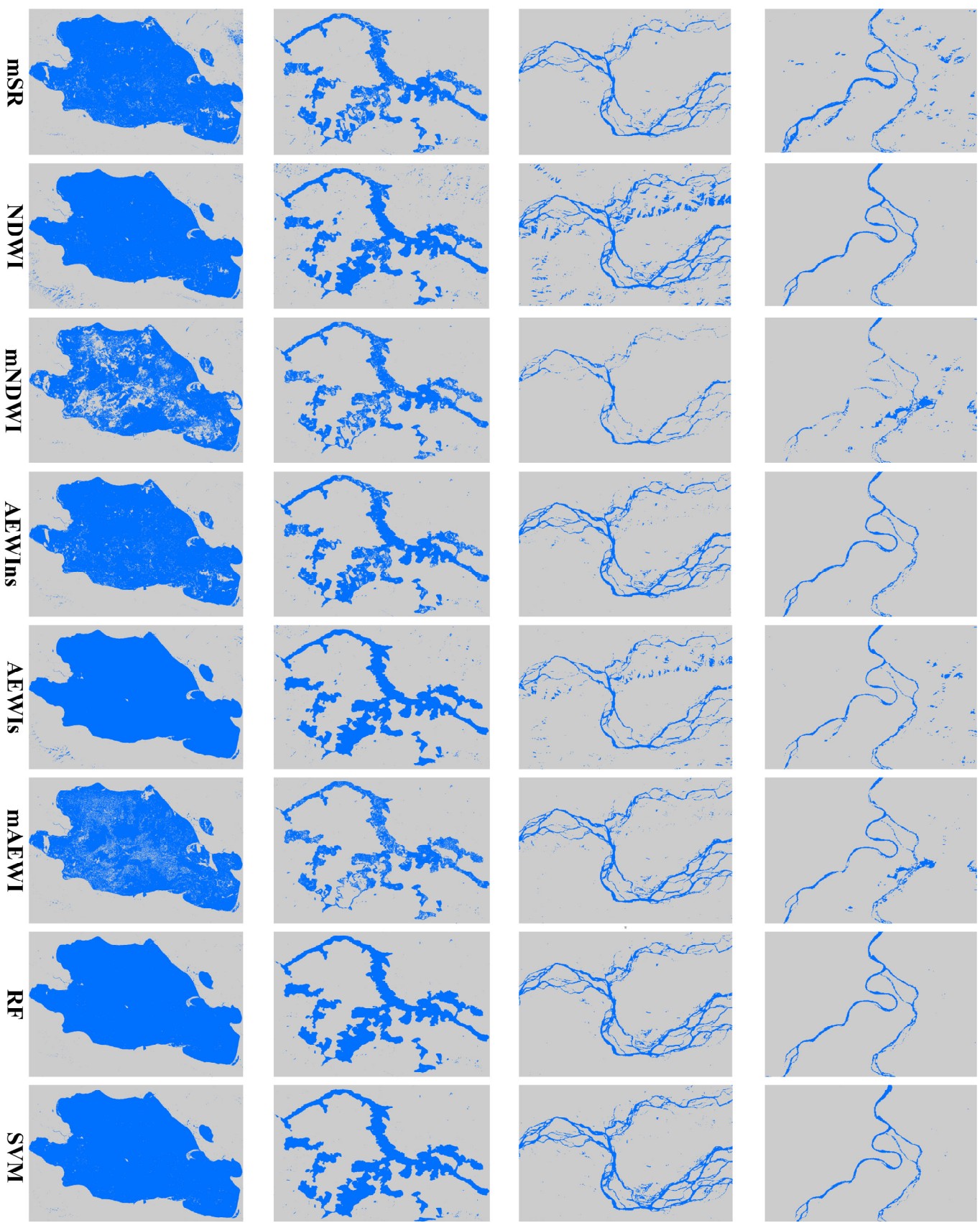

**Figure 7.** Comparison of the recognition accuracy of different water extraction algorithms for rivers and lakes. L9C2_SR imageries were synthesized in false color in bands 5, 4, and 3, and stretched with 2.5 standard deviations. The white oval circles on the image are the regions where the recognition

abnormality may occur, which are A1, A2, A3, A4, A5, A6, A7, A8, and A9 from left to right. The coordinates of the central points of the four images are (100°10′29.0″E, 36°54′6.9″N), (90°44′44.1″E, 28°59′22.7″N), (90°45′34.8″E, 29°20′5.3″N), and (102°27′50.5″E, 33°27′20.8″N) from left to right. Regions colored in blue are waterbody areas, and gray are non-waterbodies.

### 4.3. Analysis of Lake Waterbody Area Extraction Results

The non-lake waterbody dataset produced the lake waterbody dataset, and the SRTM DEM had this research. The former was constructed in Section 3.1 and was used to eliminate the non-lake waterbodies in the extraction results [68], and the latter was used to correct the shadow influence [44]. Referring to the classification of lake waterbody, the area distribution of lake waterbody in the QTP region the resolution of lake waterbody extraction in the previous study [3], the lake waterbody area in this study is divided into 30–100 km$^2$ (type 1), 100–500 km$^2$ (type 2), and over 500 km$^2$ (type 3). As shown in Table 4, we randomly selected ten lakes in each category to build a lake waterbody area validation set to quantitatively analyze the extraction results of the lake waterbody area to measure the accuracy of extracting the lake waterbody area of three types by different algorithms.

**Table 4.** Extraction results of lake waterbody area by different models.

| ID | Central Location (°) | Area/km$^2$ | Type | ID | Central Location/° | Area/km$^2$ | Type |
|----|----------------------|-------------|------|----|--------------------|-------------|------|
| 01 | 89.4541E, 32.3388N | 34.7176 | 1 | 16 | 83.0601E, 35.2735N | 246.6113 | 2 |
| 02 | 90.5172E, 28.9495N | 38.6561 | 1 | 17 | 88.7222E, 31.5855N | 268.9577 | 2 |
| 03 | 88.6920E, 32.3169N | 44.1437 | 1 | 18 | 90.1922E, 35.7528N | 294.7249 | 2 |
| 04 | 82.3336E, 31.6280N | 54.4298 | 1 | 19 | 89.4425E, 36.3302N | 386.9375 | 2 |
| 05 | 86.2695E, 35.2978N | 62.2600 | 1 | 20 | 73.4069E, 39.0290N | 413.4563 | 2 |
| 06 | 88.1369E, 36.1962N | 65.5249 | 1 | 21 | 88.9544E, 34.5827N | 511.8270 | 3 |
| 07 | 86.7384E, 31.5679N | 67.8710 | 1 | 22 | 97.2666E, 34.9309N | 549.8463 | 3 |
| 08 | 85.2322E, 31.5679N | 71.8135 | 1 | 23 | 88.2833E, 31.1572N | 555.8818 | 3 |
| 09 | 85.8104E, 33.6649N | 73.8108 | 1 | 24 | 88.3989E, 37.0775N | 599.5535 | 3 |
| 10 | 95.8114E, 36.7416N | 80.8892 | 1 | 25 | 97.5895E, 38.2928N | 639.1734 | 3 |
| 11 | 91.1596E, 31.7089N | 148.5164 | 2 | 26 | 97.7020E, 34.9062N | 656.0598 | 3 |
| 12 | 87.1744E, 34.5513N | 167.8379 | 2 | 27 | 90.4774E, 34.7984N | 693.1980 | 3 |
| 13 | 89.9783E, 32.4493N | 188.9746 | 2 | 28 | 85.6116E, 30.9289N | 1048.9105 | 3 |
| 14 | 92.1340E, 35.2207N | 203.0216 | 2 | 29 | 90.0623E, 33.4379N | 1137.2408 | 3 |
| 15 | 84.5659 E, 35.4053N | 221.7653 | 2 | 30 | 100.1977E, 36.8884N | 4538.2366 | 3 |

Figure 8 shows the error analysis results of lake waterbody extraction. Among all algorithms, the accuracy of the lake waterbody area extracted by the RF model is the highest (1.505%), followed by the SVM model (1.624%). In the threshold segmentation algorithm, the NDWI (3.501%) is the most accurate model, followed by AEWIs (6.789%). Comparing the area extraction accuracy of different types of lakes, we found that in almost all models, the error of the large lakes is the largest and that of the medium lakes is the smallest. The reasons for the significant lake errors are analyzed in combination with the identification results of Qinghai Lake (the largest lake in the QTP) in Figure 7. The errors mostly appear in the dark areas within the lakes. During the experiment, we also found that the dark regions mainly appear in large lakes with larger surfaces and deeper depths. This is because deep water areas of large lakes are more prone to hyperreflective and dark spots, while larger spaces are also vulnerable to lake ice and shadowing, leading to the emergence of the dark regions. Therefore, the identification error is more significant for large lakes. In addition, the spatial resolution of L9C2_SR data is 30 m. When identifying small lakes, it is easy to generate commission errors at the mixed pixels of the waterway junction. Therefore, error analysis of small lakes is greatly influenced by error classification, resulting in large error fluctuations.

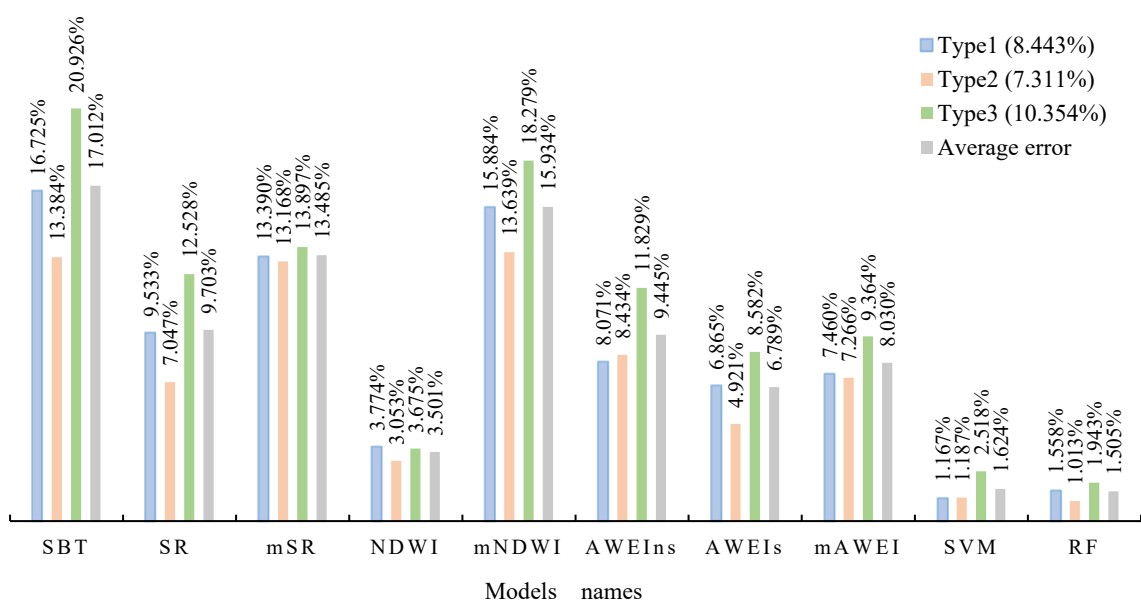

**Figure 8.** The error analysis (See Section 3.4 for error analysis formula) of different waterbody extraction algorithms for various types of lake area extraction. The X-axis of the histogram is the model's name, and the percent labels are the average extraction error of different lake areas under various models. The percent labels behind the legend are each algorithm's average lake waterbody area extraction error.

### 4.4. Analysis of Comparison with Landsat-8 Data

The Landsat-9 image attracted much attention in the field of plateau water and lake extraction because it has almost the same band setting as the widely recognized Landsat-8 image, and they have good consistency in theory. Therefore, they can complement each other and improve the time resolution of data. To verify the consistency and availability of Landsat-9 and Landsat-8 data, we selected an area of more than 24,000 square kilometers in the study area for comparative study.

The imaging time is controlled in May and June to reduce the impact of environmental transformation on data. The cloud removal and mosaic of data were carried out according to the above process, and the images of Landsat-8 and Landsat-9 were obtained, as shown in Figure 9a,b. It can be seen that Landsat-8 data have higher cloud coverage in this area (as shown in the circled area in Figure 9a) due to the differences in track settings and shooting time, and Landsat-9 data can complement it. After eliminating the missing and abnormal pixels, the reflectance and NDVI of Landsat-9 and Landsat-8 data were linearly regressed, and the results are shown in Figure 9c,d. It can be seen that the linear regression coefficients of reflectance and NDVI of the two data tend to be 1, and R is also greater than 0.91, which shows that they have good consistency and can be used jointly for the extraction of water and lake areas.

Then, the RF and NDWI algorithms, which performed best in machine learning and threshold extraction algorithms, were selected for water extraction and accuracy verification. According to the above technical process, the accuracy of the two data is shown in Table 5 below. Firstly, due to the smaller classification range and the reduced complexity of the environment, each algorithm's recognition accuracy and kappa coefficient are improved compared to the extraction of the whole plateau. In addition, it can be seen that the maximum differences in the OA and kappa coefficient of the two images are 1.1% and 0.025, respectively. This shows that Landsat-9, similar to Landsat-8, can recognize plateau waterbodies well. At the same time, it is also proven that although the radiation resolution of Landsat-9 is improved to 14 bits, it has no apparent advantage in relatively simple tasks, such as waterbody recognition.

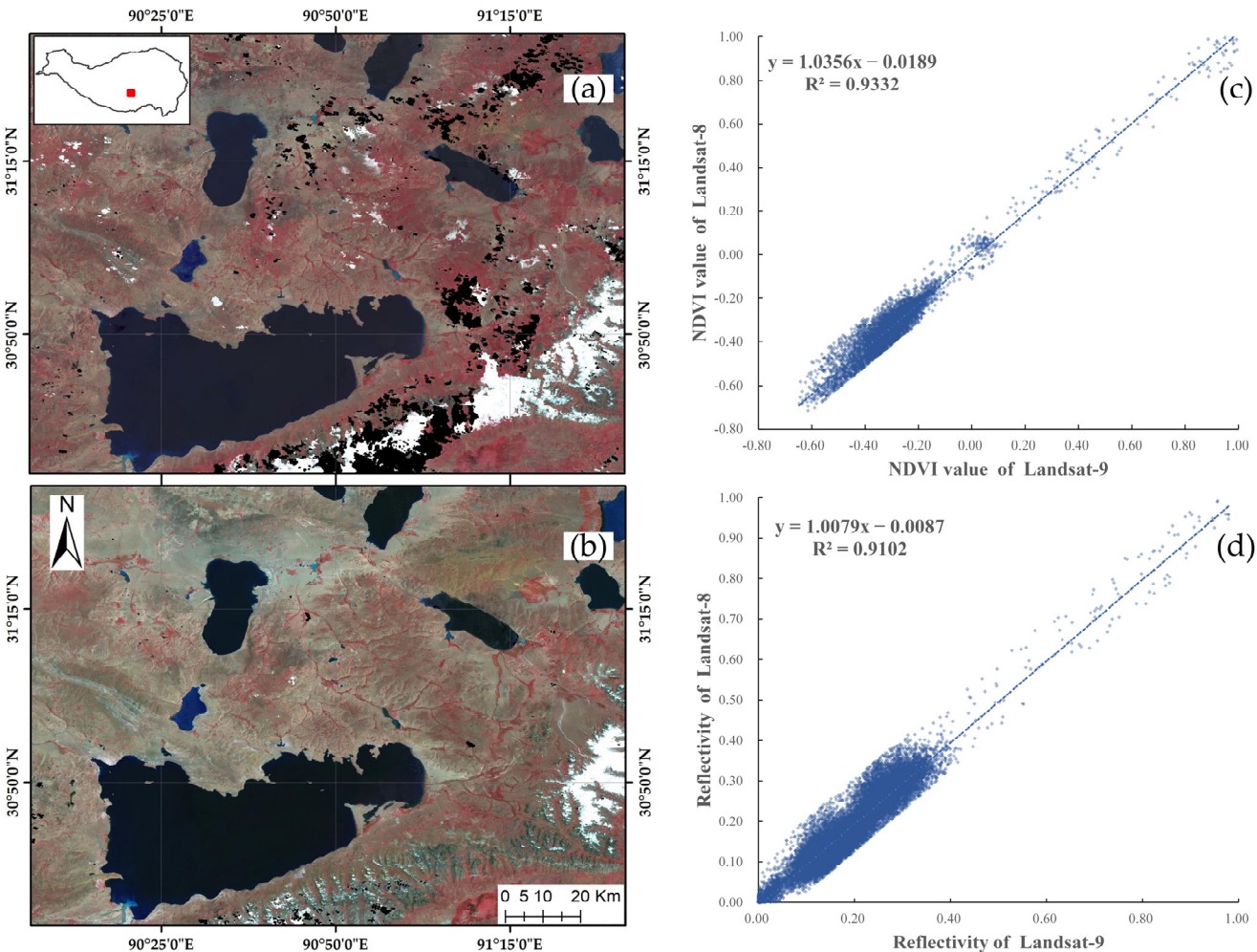

**Figure 9.** Image acquisition and linear regression. (**a**) Landsat-8 image; (**b**) Landsat-9 image; (**c**) linear regression of NDVI; and (**d**) linear regression of reflectance.

**Table 5.** Comparison of the overall accuracy of water extraction.

| Mode | Indicator | Landsat-8 | Landsat-9 | Difference * |
|------|-----------|-----------|-----------|--------------|
| RF | OA(%) | 97.92 | 96.82 | 1.10 |
| | Kappa coefficient | 0.922 | 0.947 | −0.025 |
| NDVI | OA(%) | 92.27 | 92.95 | −0.68 |
| | Kappa coefficient | 0.857 | 0.871 | −0.014 |

* The difference refers to the OA/kappa coefficient of Landsat-8 minus the OA/kappa coefficient of Landsat-9.

## 5. Conclusions

It is of great significance for global climate change to quickly and accurately obtain information on the changes of lakes in the QTP based on remote sensing technology, but there is a specific relationship between the performance of different lake waterbody extraction algorithms and application scenarios with remote sensing data. To explore the interaction between Landsat-9 data and the accuracy of varying algorithm models for QTP lake waterbody extraction and determine the algorithm for large-scale QTP lake waterbody area extraction suitable for Landsat-9 imagery, this study selected 10 models widely used in waterbody extraction, carried out comparative research leveraging the GEE platform, and found the following conclusions:

Affected by clouds and shadows, the Landsat-9 data with limited quality, and only 30 m resolution, the waterbody extraction model still achieved the best 96.59% overall

accuracy and 1.505% average error in the extraction of QTP waterbody and lake waterbody area extraction. It is proven that those algorithms can effectively extract and identify QTP waterbodies using Landsat-9 data. With the publication of 15 m panchromatic data and the acquisition of high-quality images, there is still more significant improvement in the accuracy of waterbody and lake waterbody area extraction. Moreover, compared with the threshold segmentation algorithms, the machine learning algorithms have significant advantages in extracting large-scale QTP waterbody and lake waterbody areas. Comparing the two machine learning algorithms, under equal overall accuracy and average error, the operation efficiency of RF on the GEE is significantly higher than that of the SVM classifier. Therefore, the RF algorithm is more recommended in similar studies. With the development of the GEE platform, the constraint of computing power on model selection will be greatly reduced. More models with complex calculations but higher accuracy can be considered in future related research. Finally, among the traditional threshold segmentation waterbody extraction algorithms, the best extraction result is the NDWI method. The overall accuracy of waterbody extraction is 89.89%, and the average lake waterbody area extraction error is 3.501%. The NDWI method is a recommended practice in scenarios with limited samples or high operational efficiency.

This study also has some areas that need further research and improvement. First, because Landsat-9 data were just released, the data used in the study are mainly concentrated in January–April. During this period, there was still a part of incomplete melting lake ice in the lakes of the QTP, which makes the radiation transmission of waterbodies more complex and affects the accuracy of waterbody identification. However, due to limited data sources and relatively complex experiments, this study did not explore and eliminate the impact of lake ice. Secondly, to ensure the complete cloudless coverage of the whole study area as much as possible in the case of limited images, we used the min () function for image synthesis, which enhanced the influence of shadows on the image. Finally, the optimal thresholds for different underlying surfaces and scales are different for large-scale water extraction. Therefore, dividing the study area into different scale zones for threshold selection can further improve the accuracy of waterbody recognition. In the follow-up, we will further research the above problems and deficiencies when the data and computing power is improved.

**Author Contributions:** Conceptualization, X.L. and Y.Z.; methodology, D.Z.; software, C.J.; investigation, Y.S. and D.L.; data curation, K.Q.; writing—original draft preparation, X.L. and C.J.; writing—review and editing, Y.L. and D.C.; visualization, S.L.; supervision, D.Z.; project administration, Y.Z.; funding acquisition, H.L. All authors have read and agreed to the published version of the manuscript.

**Funding:** This research was funded by Major Science and Technology Projects in Anhui Province (NO. 202003a06020002), Key Research and Development Projects of Anhui Province (NO. 2021003), Special Support Plan for High-Level Talents of Anhui Province (NO. 2019), Major science and technology projects of high-resolution earth observation system (76-Y50G14-0038-22/23), Natural Science Research Project of Universities in Anhui Province (KJ2021A1063), Science and Technology Plan Project of Chuzhou City (2021ZD013) and Wuhu key R & D plan (2020ms1-3).

**Data Availability Statement:** The data presented in this study are available on request from the corresponding author.

**Acknowledgments:** We are also thankful to all anonymous reviewers for their constructive comments provided on the study.

**Conflicts of Interest:** The authors declare no conflict of interest.

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
