# Peer review of "Comparison of Lake Area Extraction Algorithms in Qinghai Tibet Plateau Leveraging Google Earth Engine and Landsat-9 Data"

_remotesensing, doi:10.3390/rs14184612_

Round 1
Reviewer 1 Report
See review in attached PDF.

Author Response
Dear Reviewer,
Special thanks to you for your valuable comments and suggestions. Those comments are valuable and very helpful. We have read through comments carefully and have made corrections. Based on your submissions and suggestions, we revised and uploaded the manuscript. The manuscript is uploaded in both Word and PDF versions, of which Word uses the "Track Changes" function to facilitate your understanding of the changes to this manuscript, and the PDF version is a full version without traces to make it easier for you to read the manuscript.
We would love to thank you again for allowing us to resubmit a revised copy of the manuscript and we highly appreciate your time and consideration. All your comments and questions have been answered one by one in the responses below. If you have any questions, please feel free to contact me. We sincerely look forward to communicating with you.
Best regards,
Mr. Xusheng Li

Reviewer 2 Report
This paper conducts a methodological comparison study of water extraction in the Tibetan Plateau region for the newly launched Landsat9 satellite data, which is significant for enriching the data of lake changes in the Tibetan Plateau region and exploring climate change and ecological environmental protection in the Tibetan Plateau. Compared to Landsat-8, satellite sensor in Landsat-9 has some improvements but still has some similarities in design. In this study, only the accuracy of nine models for extracting water bodies in Landsat9 images was investigated, and the performance was not compared with that of Landsat8 images.
Moreover, in the results of the study, although the extraction accuracy of each model is illustrated, the comparative analysis of the model's time consumption and resource usage is not performed, which cannot fully reflect the advantages that machine learning methods such as SVM and RF have over water body index models such as NDWI in water body extraction. The study is generally innovative, and it is recommended to add the necessary comparative analysis of model time consumption and resource usage.
In models such as the water body index method, the authors manually adjust the thresholds based on lake sample point data (L312) So, why not use the zonal automatic threshold determination method? Although the authors made certain explanations, such as the lake area of the Qinghai-Tibet Plateau accounts for a relatively small area compared to the Qinghai-Tibet Plateau as a whole, which affects the use of automatic extraction methods such as OSTU. However, a large number of studies have now shown that the thresholds are different in different regions, and the authors can use zoning to automatically determine the thresholds to start the study and reduce human errors.
In terms of result validation (Section 4.3), the authors used TPLA_V3 data as the true value for lake area data accuracy validation, which is inappropriate. The seasonal and interannual variability of lakes on the Tibetan plateau is significant, especially for small lakes.The authors used the lake interpretation results before 2021 as true values to verify the lake interpretation results in 2022, which obviously introduced unnecessary errors. It is suggested that the authors decode the February-April 2022 lake data of Landsat9 or Landsat8 as the true values by manual decoding. Otherwise, the authors' current accuracy comparison is wrong.
<General Comments>
- More suitable title should be selected for the article.
- Abstract needs to modify: the abstract should contain Objectives, Methods/Analysis, Findings, and Novelty /Improvement.
- The necessity and innovation of the article should be presented to the introduction.
- It is suggested to present the structure of the article at the end of the introduction.
- It is suggested to compare the results of the present research with some similar studies which is done before.
- The major defect of this study is the debate or Argument is not clear stated in the introduction session. Hence, the contribution is weak in this manuscript. I would suggest the author to enhance your theoretical discussion and arrives your debate or argument.
<Specific comments>
- L152-153:It is recommended to list all nine countries.
- L157: km2 superscript error.
- L239 he is changed to the.
- L565 modeals is changed to models.
Author Response

(The authors gave the same response as above.)

Round 2
Reviewer 2 Report
My comments have been very well responded to by the author. Now that I see that the manuscript has been well refined, I would recommend it for publication.
Author Response
Dear Reviewer,
First of all, thanks very much for your professional and careful review. Your constructive suggestions have played a positive role in improving the level of our manuscript. Secondly, thank you again for your approval of our revised manuscript, and we highly approve your time and consideration. If you have any questions, please feel free to contact me. We sincerely look forward to communicating with you.
Warmest Regards,
Mr. Xusheng Li